# Design and evaluation of a prototype medical device for robotic and manual percutaneous dilatational tracheostomy

Yuan Tang[1,2], Glen Cooper[2], Damian Crosby[1,2], Brendan A. McGrath[3], Andrew Weightman[1,2]*

1 Manchester Centre for Robotics and AI, University of Manchester, Manchester, United Kingdom, 2 Department of Mechanical and Aerospace Engineering, University of Manchester, Manchester, United Kingdom, 3 Manchester University Foundation Trust Wythenshawe Hospital Intensive Care Unit, Manchester, United Kingdom

* andrew.weightman@manchester.ac.uk

## Abstract

Percutaneous Dilatational Tracheostomy (PDT) is frequently performed at the intensive care unit bedside. Perioperative movements of instruments, especially inserting and removing the needle and dilators, may cause severe complications in around 10% of total patients. This research aims to reduce the complications associated with PDT instruments and simplify the procedure by proposing a prototype medical device called TrachyPen. Either used robotically or manually, TrachyPen combined the PDT puncture and dilation into a single step to eliminate the use of multiple instruments and their complex operations. Considering the insertion and dilation force as a critical safety factor for PDT, fifty simulated PDT insertion and dilation experiments were conducted on porcine back skins to evaluate the force profiles of TrachyPen and compare them with the Ciaglia Blue Rhino (CBR) dilator, commonly used in current clinical practice. Results illustrated a smaller stoma of 13.67(164) mm and a smaller insertion force of 53.45(807) N by TrachyPen compared with CBR (16.17(205) mm and 69.01(842) N). This research proves the concept of the TrachyPen design and the potential of using it for robotic PDT puncture and dilation.

## Introduction

Tracheostomy is a frequently performed technique creating an alternative airway in the front of the neck, typically to manage obstruction or to facilitate prolonged mechanical ventilation [1]. Around 20,000 tracheostomy procedures are performed annually in the United Kingdom [2,3]. Methods for performing tracheostomy include the surgical tracheostomy (ST) and Percutaneous Dilatational Tracheostomy (PDT) [4]. Compared with surgical tracheostomy, PDT has similar complication rates of around 10% but shorter hospital stays of up to 4 days since it can be completed at the intensive care unit bedside [5], which makes PDT an efficient and preferred alternative for patients with normal anatomy [6,7].

**Data availability statement:** All relevant data are within the manuscript and its Supporting information files.

**Funding:** The author(s) received no specific funding for this work.

**Competing interests:** The authors have declared that no competing interests exist.

However, PDT may still be associated with all of the complications of ST, such as major bleeding, pneumothorax and tracheal wall injury [8–10]. During the initial needle puncture process, the oesophagus and posterior tracheal wall may be injured by excessive PDT puncture, in which the needle movement does not stop in time after penetrating through the pretracheal tissue [10,11]. During PDT dilation, excessive dilation insertion could lead to tracheal ring collapse and posterior tracheal wall damage, and the guidewire may be kinked due to the dilator direction being too tangential [12]. Some complications may result from the manipulation of PDT instruments. Tracheal wall perforation and tracheal ring collapse may be attributed to excessive dilating force. The maximum allowable force threshold for PDT dilation is in the range of 31.6 N to 87.7 N [9,13]. Furthermore, the complexity of manipulating instruments and multiple steps in PDT may contribute to the long learning curve of PDT quantified from 20 and maybe up to 70 procedures [14,15] and associated complications. Due to the patient's underlying critical illness, these effects can rapidly become life-threatening.

The instruments used for PDT procedures include a hollow needle, a guidewire, and different types of dilators which are used to expand the stoma so that it can accommodate the tracheostomy tube. The most commonly used dilators in the UK [16] are Ciaglia Blue Rhino (CBR) [17] (Cook Critical Care, Limerick, Ireland), Grigg's Guidewire Dilating Forceps (GWDF) (ICU Medical Inc., CA, USA), [18,19], PercuTwist [20] (Teleflex, NC, USA) and Ciaglia multiple-dilator set [17] (Cook Critical Care, Limerick, Ireland). The mechanical structure and corresponding operations of all dilators may bring extra risks. Inspiration for introducing the PercuTwist technique (expand the tissue through a screw-like dilator) came from the high resistive force while pushing CBR into the trachea [21]. Although GWDF usually does not require a large pushing force, under-dilation or over-dilation may occur while using GWDF because of the inaccurate manipulation of forceps [22]. Using PercuTwist for dilation may lead to high pressure inevitably exerted on the rigid anterior tracheal arch and compressed upon the cervical block, which could result in tracheal wall injury during all antegrade dilation processes [23].

Typical PDT procedures are performed as follows [24]:

1. Transcutaneous needle puncture into the trachea.
2. Guidewire insertion through the hollow needle, needle removal.
3. A. Tapered single blunt dilator insertion. B. Insertion of progressively larger separate dilators.
4. Dilator removal and tracheostomy tube insertion.

Ventilation of the patient continues throughout the procedure but ventilation is partially obstructed during the procedure itself. The puncture and dilation process should be completed promptly to resume lung ventilation (via the tracheostomy tube) as soon as possible. Excessive procedure time could increase the risk of hypoxia during the procedure [25,26]. PDT procedures in current clinical practice require insertions and removals of multiple instruments to complete the PDT, which can prolong the procedure time.

Considering the aforementioned limitations associated with current PDT instruments, this research aims to design and evaluate a new medical device for performing PDT through the following objectives:

• Reduce tissue damage caused by the mechanical puncture and manipulation of current PDT instruments.
• Simplify the PDT and shorten its duration by combining multiple steps into a single procedure with a single device.

In this paper, we proposed a new prototype medical device called TrachyPen to perform PDT based on our previously published works about design requirements [27,28]. TrachyPen combined the functions of the needle and the dilator by performing PDT puncture and dilation within a single insertion. TrachyPen was designed for robotic PDT procedure, as shown in Fig 1, but it can also be held by hand and operated manually or with motor assistance. TrachyPen could facilitate semi-autonomous PDT in the future, in which the needle puncture and dilation are performed by robots to simplify the procedure and reduce risks associated with inaccurate human operations.

## Materials and methods

### A. Mechanical design of TrachyPen

**Design specifications and preliminary design.** To identify the clinical needs and provide an appropriate solution, the mechanical design of TrachyPen was guided by the design requirements developed from a literature review and an online questionnaire for healthcare professionals to capture their perspectives. In our previous work, twenty-one healthcare professionals with a mean working experience of PDT of 11.3 years completed the questionnaire [27]. Based on the literature review and questionnaire results, a development framework for PDT instruments was established to guide the conceptual design. The design requirements to guide the mechanical design are summarized in Table 1. Details of the design requirements and conceptual design are described in our previous works [27,28].

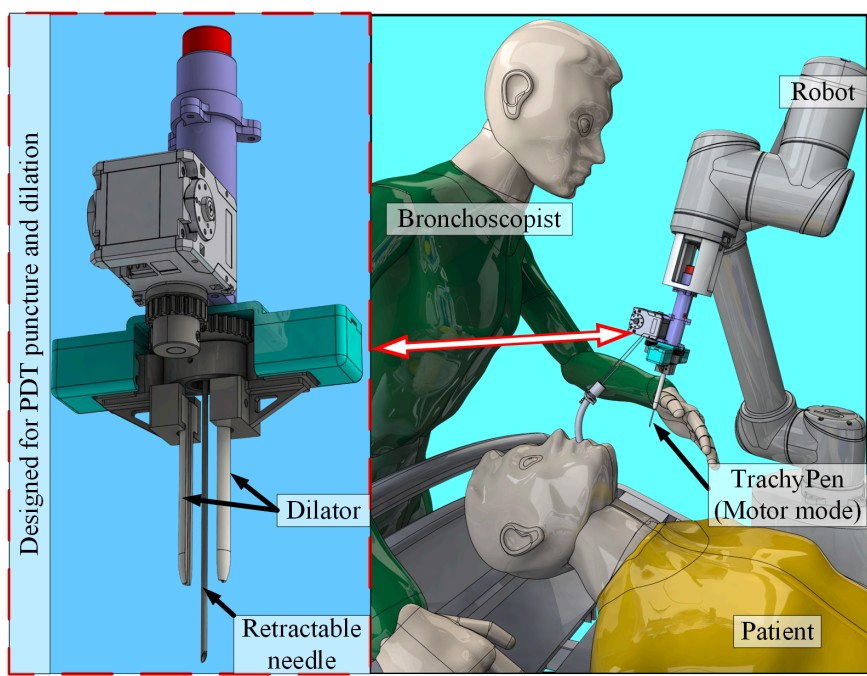

**Fig 1**. **The scenario of using TrachyPen to perform the PDT procedure robotically.**

**Table 1**. Potential improvements of current PDT and device design requirements for a puncture and dilation device [28].

| Potential improvements of current PDT | Frequency[a] |
|---|---|
| Guarantee accurate PDT placements | 14/21 (66.7%) |
| No interference with ventilation | 6/21 (28.6%) |
| Minimize bleeding | 5/21 (23.8%) |
| Safe dilation | 4/21 (19.0%) |
| Integrate into current workflows | 2/21 (9.5%) |
| **Device design requirements** | **Frequency[a]** |
| Easy to use | 13/21 (61.9%) |
| Compact size | 8/21 (38.1%) |
| Cost-effective | 7/21 (33.3%) |
| Short procedure time | 5/21 (23.8%) |
| Fit various patient anatomy | 4/21 (19.0%) |
| Reliable | 2/21 (9.5%) |
| Easy to clean | 1/21 (4.8%) |

[a]The frequency of mentioning is calculated as (number of participants mentioned the term/all participants)

## B. Working principle of TrachyPen.

The working principle of TrachyPen has been validated during the conceptual design stage in our previous work [28]. Illustrated in Fig 2, a PDT can be completed using TrachyPen through four steps:

1. Needle Puncture: Perform the puncture with dilator arms locked in the closed position and the needle extended. Appropriate lengths of dilator arms, needles, and sliding slots are selected according to the patient's pretracheal tissue thickness, which is estimated using preprocedural ultrasound imaging [29]. The puncture is confirmed by visualizing the needle tip inside the trachea through bronchoscope images.
2. Guidewire insertion and blunt dilation: Insert the guidewire through TrachyPen and retract the needle to prevent damage to the posterior tracheal wall. Then push the TrachyPen body for blunt dilation until the dilator arm tip reaches inside the trachea and is visualized by a bronchoscope.
3. Dilation: Align the sliding rail with the neck, then perform the dilation either by rotating the handle manually or using a motor. Repeat the dilation movement until the resistance from the tissue significantly reduces.
4. TrachyPen removal and manual tube insertion: Remove TrachyPen carefully without bringing the guidewire outside and insert the tracheostomy tube manually.

The main feature of using TrachyPen is that only one insertion is required for the whole procedure.

**Mechanical design details of TrachyPen.** The revised design of TrachyPen based on the clinical feedback is shown in Fig 3. The name comes from its retractable needle mechanism, which was inspired by a ball-point pen with retractable infill. The core principle of TrachyPen is to combine several PDT steps and use a single device. Compared with traditional PDT instruments, which need insertions and removals separately, the potential advantages of TrachyPen are:

- Reduce the steps needed for PDT (i.e., number of insertions and removals of PDT instruments) and the number of instruments used with simplified puncture and dilation mechanisms. This could potentially reduce the risks associated with the PDT instruments, the complexity of operating multiple instruments, and the procedure time.
- The reusable TrachyPen could potentially reduce the instrument costs compared with traditional single-use PDT kits.

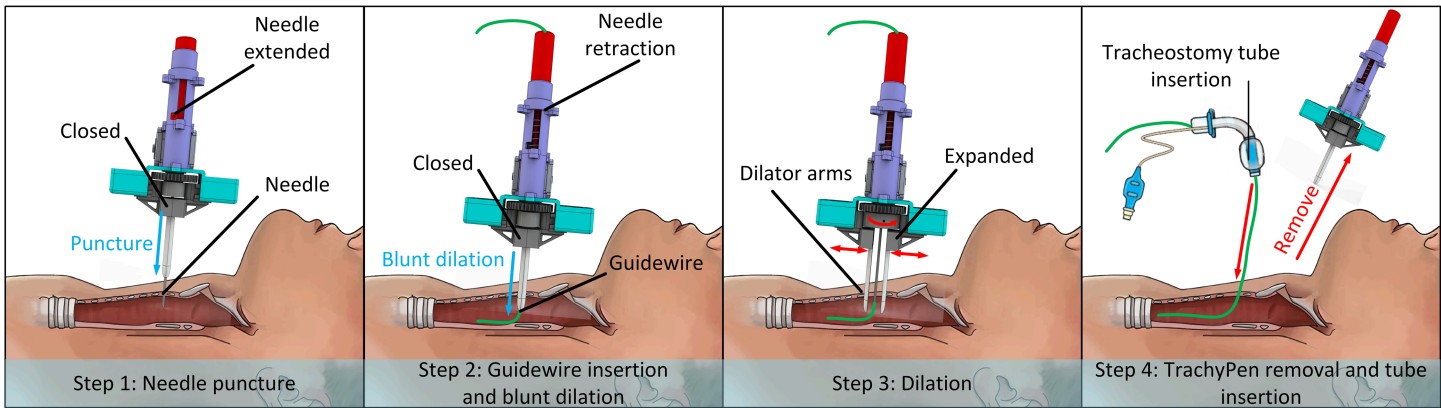

**Fig 2**. **Four steps of using TrachyPen to complete the PDT procedure.** Step 1: puncture; Step 2: needle retraction and guidewire insertion; Step 3: dilation; Step 4: TrachyPen disengage and manual tube insertion.

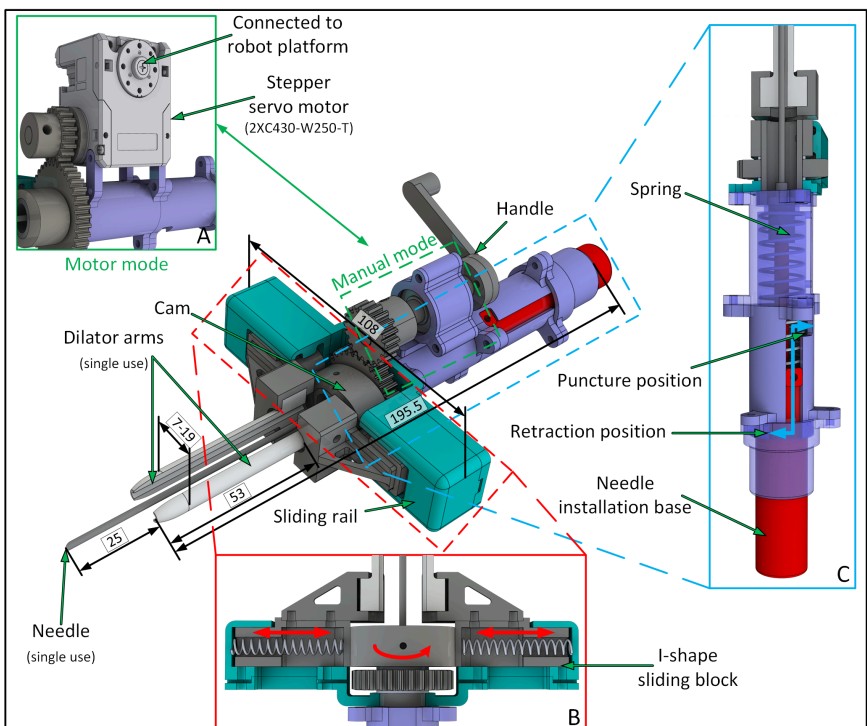

**Fig 3**. **Overview of the mechanical design of TrachyPen.** (A) TrachyPen can be used by rotating the handle manually or with motor assistance. (B) The stoma created by the puncture is expanded by the sliding movement of two dilator arms controlled by a cam. Dilator arms remain closed by two compressed springs. (C) The needle is extended through an S-shape slot for puncture and retracted during dilation.

The device consists of a retractable needle, two dilator arms to expand the stoma, and a transmission chain to actuate the dilation. The device can be operated in both 'automatic' and 'manual' mode. In automatic mode, as illustrated in Fig 3(A), a stepper servo motor DYNAMIXEL 2XC430-W250-T (ROBOTIS, INC., Lake Forest, CA, United States) is installed on the TrachyPen body to control the movement of dilator arms. This is designed for robotic use where the TrachyPen body is mounted on the robot end-effector, but it is also suitable for applying dilation force when the TrachyPen is

held by hand during a manual PDT. In manual mode, the motor is replaced with a handle for manual rotation. In this case, the operator holds the TrachyPen body with one hand and rotates the handle with the other hand.

The dilator arms expand the stoma created by the needle puncture through translational movements of the dilator arms, which is similar to GWDF. The translation movement is converted from the rotation of the input shaft through a pair of gears and an elliptical cam. Illustrated in Fig 3(B), springs between the sliding blocks and sliding rails provide a force to push the dilator arms together so that they can be kept closed when not in use or during the insertion. The diameter of the dilator arms in the close mode was 7 mm, the same as the blunt dilator used in the PDT kit (Cook Critical Care, Limerick, Ireland) for the CBR technique [17,30]. The maximum dilation dimension was 19 mm, which is 3 mm larger than a CBR dilator. For various patients with different neck sizes, dilator arms with different lengths can be installed to fit the corresponding pretracheal tissue thickness. The required torque $\tau$ in N m at the input shaft was computed from the dilation force of GWDF [13,31] plus the spring force using the following equation:

$$\tau = \max\left(2R\omega\left(F_s + F_d + f\right)\cos\phi\right),\tag{1}$$

where $R$ is the distance between the contact position and the cam centre in m, $\omega$ is the gear ratio, $F_f$ is the contact friction in N and $\phi$ is the pressure angle of the cam in rad. Describe the elliptical cam as the algebraic expression $\frac{x^2}{a^2} + \frac{y^2}{b^2} = 1$ where $a$ and $b$ are the major and minor axis. Given the current rotation angle of the cam $\alpha$, the contact point between the cam and the dilator arm $\begin{bmatrix} x_c & y_c \end{bmatrix}^T$ was computed as follows:

$$y_c = \sqrt{\frac{a^2 b^2}{a^2 + b^2 \tan^2 \alpha}}.$$
$$x_c = y_c \tan \alpha \tag{2}$$

Take the derivative of the algebraic cam expression and get the tangent at the contact point between cam and dilator arms $k = -\frac{b^2 x_c}{a^2 y_c}$. Then calculate the pressure angle of this contact point $\phi = \alpha - \arctan(|k|)$. Applying the dilation force $F_d$ from [13], the input torque reaches up to 1.5 N m, which is acceptable for both manual and automatic operations.

The design of the needle retraction mechanism is modified from the mechanism of a ball-point pen and is simplified for easier manufacturing. The needle movement is limited by a slot created on the TrachyPen body, and its position is locked during puncture or after retraction. The design allows the passage of the guidewire through the TrachPen body and the needle. Before starting a PDT procedure, the needle is fixed on the installation base and the sliding key is locked into the retraction slot. As shown in Fig 3(C), a compressed spring causes the key to slide upwards and lock at the upper right end of the slot during puncture, while it slides back and retracts the needle tip backwards to the dilator arms after successfully creating the stoma at the trachea to prevent any needle damage. The minimum distance between the needle tip and dilator arms when in the puncture mode is 25 mm, which is larger than the mean pretracheal tissue thickness plus the trachea radius for non-obese adult patients (8–19.5 mm) [32,33]. The needle as well as the slot length can be customized to suit various neck sizes.

To maintain satisfactory infection control, the dilator arms and TrachyPen body are reusable while the needles are single-use. The overall weight of TrachyPen for manual use is 720g, with all parts manufactured in AISI 420 stainless steel to meet the international standard ISO 7153-1:2016, which defines the materials requirements for making surgical instruments. The needle will be removed from the side of the dilator arms to prevent contaminating the nearby parts, while the TrachyPen body will be recycled and sterilized. The TrachyPen body (except dilator arms) will be covered with a sterile plastic drape during the procedure to minimize contamination.

## C. Preliminary evaluation of TrachyPen design by clinicians through questionnaire and interview

The conceptual design of TrachyPen was evaluated by considering the feasibility and potential improvements through an online questionnaire sent out to 7 healthcare professionals [28]. Once the design concepts and proposed mechanism were verified based on the questionnaire results, one TrachyPen prototype was manufactured and evaluated during a face-to-face interview with 5 clinicians (mean working experience related to PDT: 13.4 years) from Wythenshawe Hospital, Manchester, UK. The interview was conducted on June 2nd, 2024, and the recruitment period of participants started from Apr 1st to May 1st, 2024. Verbal consents of all participants were obtained before the start of the interview and witnessed by all authors. No personal information of the participants was obtained. Questions asked during the interview included clinical opinions towards the current design concept and potential improvements regarding the mechanical structure and TrachyPen usage. The clinicians felt that TrachyPen was a feasible and practical solution to perform PDT after checking the prototype. Summaries of the key themes that emerged from the interviews are listed below:

- The dilation mechanism of TrachPen eliminates the axial force during dilation, as described in the design concept. The current dilator (CBR) requires a pushing force to expand the tissue, which could lead to damage to the tracheal wall and cartilage.
- Compared to the current dilator, which generates a radial force, the dilating mechanism of Trachypen may rip the tissue since the force only goes in two opposite directions. However, which method is optimal remains undetermined and requires further evaluation.
- The design of the Trachypen should be suitable for one-hand operation, and the weight distribution of Trachypen should be balanced. The desired poses of using Trachypen could be similar to holding a pen, which could be achieved when in the automatic mode.
- The mechanical design of TrachyPen could be further simplified to reduce the number of corners and holes, which are difficult for sterilisation.
- The development process of TrachyPen should consider infection control, including how to use and sterilize the device to minimize the hazards of contamination. One possible solution is to disassemble the device for autoclaving, which requires the materials to meet international standards for surgical instruments (ISO 7153-1:2016).
- The force feedback is essential for TrachyPen, so its force characteristics, such as insertion force and dilation force, should be evaluated in the experiments.
- When using TrachyPen to perform PDT puncture, position guidance techniques should be used so that the operator knows if the device hits the vessel, follows the correct direction, or stops at the desired position.

## D. Experiments for TrachyPen evaluation

Experiments were conducted on porcine tissues to evaluate the safety and feasibility of TrachyPen during PDT. Considering the insertion and dilation force as an important factor that affects the safety level of the PDT procedure, the force characteristics of TrachyPen were determined and compared with CBR, the dilator currently used in clinical practice.

**Ethics.** According to the ethical decision tool results from the University of Manchester and NHS Health Research Authority, ethical approval was not required for conducting the experiments. All porcine tissues used in the experiments were purchased from the food chain and originally intended for human consumption. The ethical exemption letter from the University of Manchester was attached as the supplement material S1 File.

**Experiment setup and tissue sample preparation.** The experimental setup is shown in Fig 4(A)–4(E). A Franka Emika Panda robot (Franka Robotics GmbH, Munich, Germany) was used to perform simulated PDT dilations. A TrachyPen prototype was manufactured and installed on the robot, with the servo motor installed for automatic operation. TrachyPen was replaced with 3D-printed mounting flanges when testing the CBR and blunt dilator (Cook Critical Care, Limerick, Ireland). Predefined end-effector pose trajectories were applied so that the robot automatically moved to the start

   

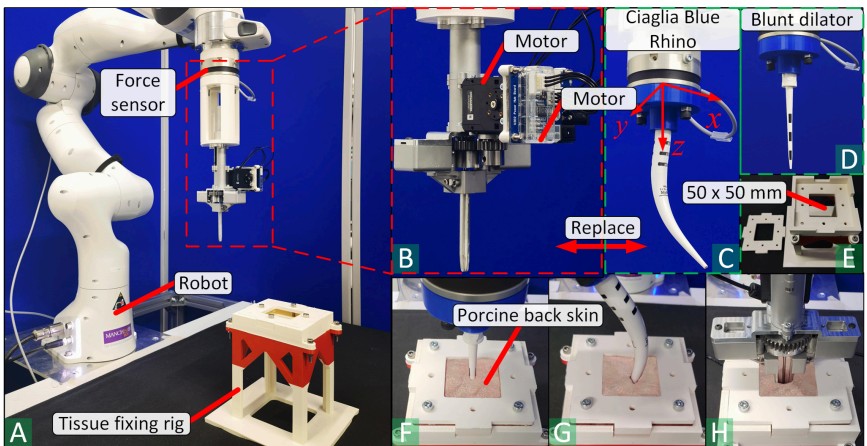

**Fig 4**. **Experiments of TrachyPen evaluation.** (A) Experimental setup. (B) Overview of the TrachyPen prototype in the automatic mode. (C-D) A Ciaglia Blue Rhino and a blunt dilator were installed on the robot to perform simulated PDT dilation. (E) The tissue fixing rig. The porcine skin samples are mechanically fixed using screws and the cap. The insertion is to be placed through the 50 × 50mm hole. (F-H) Insertions of the blunt dilator, Ciaglia Blue Rhino and TrachyPen on the porcine back skins.

position and completed the PDT dilation at the designated location. The start position for insertion for TrachyPen and CBR was defined as the position when the needle/CBR tip was about to insert into the upper surface of the porcine tissue sample vertically. The insertion force was collected using an OptoForce HEX-H sensor (OnRobot A/S, Odense, Denmark) with a precision of 0.5 N.

Porcine back skins (including surface skin, fat, and muscle layers), considered appropriate material to simulate human neck structure without excessive insertion force [34], were used for experiments. Tissue samples were purchased from local butchers (Hulme Fish and Meat Market, Manchester, M15 5JT). Samples were sliced into 4(1) mm thickness to simulate human neck skin and pretracheal tissue without applying excessive insertion force of TrachyPen/CBR.

A 3D-printed test rig (see Fig 4(E)) with a 50 × 50 mm hollow (see Fig 4(E)) was used to fix the tissue, where the insertions of instruments were performed at the hollow. Samples were fixed inside the test rig mechanically using screws and caps to prevent motion during the insertion. The tissue sample was replaced after each insertion.

**Experiment procedure and data collection.** The dilation procedure for TrachyPen is:

1. Manually make a needle puncture on the sample. The needle direction is perpendicular to the skin surface.
2. Move and align TrachyPen with the needle stoma. Insert TrachyPen along the guideline (perpendicular to the skin, defined in the robot controller) robotically using predefined trajectory points.
3. Actuate the motor to expand the stoma twice (dilator arms opening and closing twice), then rotate the TrachyPen 90° around the needle direction and repeat the dilation twice (four dilations in total).

The dilation procedure for CBR is:

1. Manually make a needle puncture on the tissue. The needle direction is perpendicular to the tissue surface.
2. Align the needle stoma with the blunt dilator and perform the blunt dilation along the guideline (perpendicular to the skin, defined in the robot controller) robotically using predefined trajectory points.
3. Align the stoma with the CBR tip and perform the dilation robotically using predefined trajectory points. The trajectory is a curve making the CBR segment, which is about to enter the stoma perpendicular to the skin.

Insertion and dilation force data applied with the TrachyPen, blunt dilator, and CBR were collected. Both the force along *x* and *z* (see Fig 4(C)) were recorded for CBR, while only the force along *z* was recorded for TrachyPen and blunt dilator. For each instrument, the maximum insertion force along *z* during the whole insertion process was recorded. Since there is no sensor to directly measure the dilation force of TrachyPen, the dilation force of TrachyPen (force applied on the dilator arms) was calculated using Eqs (1) and 2, given the cam angle and load readings from the motor.

After performing the PDT dilation, the tissue thickness and stoma length were measured using a calliper. The stoma length was measured at the skin surface immediately after removing the dilator without changing its shape or pressing the surroundings. The shape of the stoma was observed. If it were an ellipse, the stoma length would be defined as the diameter of the longer axis. If the shape was irregular (cross or triangle shape), the stoma length was defined as the diameter of its minimum circumscribed circle.

Data were visualized and analysed using SPSS 24 (IBM, New York, USA). The distributions of force and length data were examined with Shapiro-Wilk tests of normality. Independent t-tests were performed for stoma length and insertion force data to identify if significant differences exist between the CBR and TrachyPen groups. Summary statistics were reported as mean $\pm$ standard deviation (range). Statistical significance was accepted at the P < 0.05 level.

## Results

In total, 25 TrachyPen and 25 CBR dilations were successfully performed on the porcine back skins. The Shapiro-Wilk normality tests confirmed that all results were normally distributed. The mean sample thicknesses for CBR and TrachyPen groups were 4.18(61) mm (3.06 mm to 5.00 mm) and 3.98(51) mm (3.17 mm to 5.04 mm) respectively. Without any significant difference ($P = 0.207 > 0.05$) between the sample thickness of the two groups, which heavily affected the insertion force value, the acquired force data for the CBR and TrachyPen groups were considered comparable.

As expected, the mean stoma length of CBR dilation was 16.17(205) mm (11.45 mm to 18.91 mm), a mean difference of 2.50 mm larger than that of TrachyPen, 13.67(164) mm (10.58 mm to 16.13 mm, $P < 0.001$).

The insertion force data of CBR, blunt dilator, and TrachyPen during the procedure were recorded. The mean insertion force peak for blunt dilator insertion after the needle puncture was 24.35(317) N (18.20 N to 29.80 N). The mean insertion force peak of CBR in *z* direction was 69.01(842) N (53.50 N to 84.20 N), a mean difference of 15.56 N larger than that of TrachyPen, 53.45(807) N (42.90 N to 70.60 N, $P < 0.001$).

The motor output during the TrachyPen dilation was recorded. The motor output reached the maximum near the end of the first dilation. The mean motor load peaks during the four dilations were 91.4(40) % (85.2 to 99.0 %)(first dilation), 38.0(42) % (30.2 to 44.6 %)(second dilation), 32.5(40) % (26.1 to 38.8 %)(third dilation) and 26.6(35) % (21.3 to 33.8 %) (fourth dilation). A significant difference in the mean motor load peak was identified ($P < 0.001$) between the first and second dilation. The data are illustrated in Table 2, Fig 5, and the supplement material S2 Dataset.

## Discussions

Experiments on the porcine back skins simulated the PDT needle punctures and dilations using TrachyPen, which achieved a smaller mean insertion force than CBR. Since the insertion trajectories of CBR (curved path) and TrachyPen (straight path) are different, we expected the stoma length of CBR to be larger than that of TrachyPen, which matched the experiment results. We could analyze the force profiles and dilation patterns of TrachyPen and identify potential improvements regarding the mechanical design.

For CBR, the insertion force along the *x* direction was 80.84% smaller than the one along *z*, plus the latter better simulated the pushing movements made by the operator during an actual PDT. Therefore, the force along *z* was regarded as the characteristic insertion/dilation force for CBR. However, the insertion force along the *x* direction should not be ignored since the CBR insertion requires the operator to push and rotate it simultaneously so that its curved shape can be inserted

**Table 2. Experimental outcome of TrachyPen and Ciaglia Blue Rhino dilations on porcine back skins.**

| | Ciaglia Blue Rhino | TrachyPen |
|---|---|---|
| **Number of trials** | 25 | 25 |
| **Mean (Range) sample thickness [mm]** | 4.18(61) (3.06 to 5.00) | 3.98(51) (3.17 to 5.04) |
| **Independent t-test P value** | 0.207 | |
| **Mean (Range) stoma length [mm]** | 16.17(205) (11.45 to 18.91) | 13.67(164) (10.58 to 16.13) |
| **Independent t-test P value** | < 0.001 | |
| **Mean (Range) blunt insertion force peak [N]** | 24.35(317) (18.20 to 29.80) | – |
| **Mean (Range) insertion force peak [N]** | 69.01(842) (53.50 to 84.20) | 53.45(807) (42.90 to 70.60) |
| **Independent t-test P value** | < 0.001 | |
| **Mean (Range) insertion force peak in *x* direction (parallel to the skin surface) [N]** | 13.22(479) (5.40 to 23.00) | – |
| **Mean (Range) motor load peak [%]** | 91.4(40) (85.2 to 99.0) (first dilation) | |
| | 38.0(42) (30.2 to 44.6) (second dilation) | |
| | 32.5(40) (26.1 to 38.8) (third dilation) | |
| | 26.6(35) (21.3 to 33.8) (fourth dilation) | |
| **Calculated mean (range) TrachyPen dilation force peak [N]** | 94.69(415) (88.28 to 102.56) (first dilation) | |
| | 39.37(436) (31.30 to 46.18) (second dilation) | |
| | 33.71(413) (27.05 to 40.21) (third dilation) | |
| | 27.54(367) (22.02 to 35.00) (fourth dilation) | |

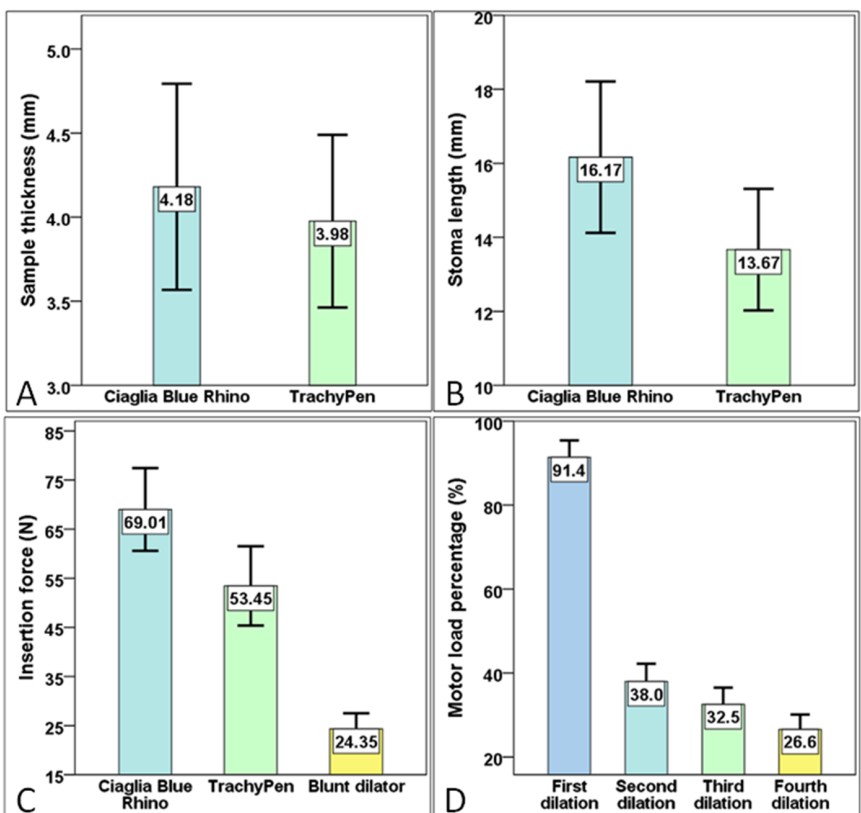

**Fig 5. Results and data distributions of simulated TrachyPen CBR dilation on porcine back skins.** (A) Sample thickness of Ciaglia Blue Rhino and TrachyPen groups. (B) Stoma length of Ciaglia Blue Rhino and TrachyPen groups. (C) Insertion force peak of Ciaglia Blue Rhino, TrachyPen and blunt dilator. (D) Motor output during TrachyPen dilation.

into the trachea. During the CBR insertion, the pushing force gradually increased along with the diameter of the CBR segment entering the stoma. The force peak was identified when the CBR segment had the largest diameter and the insertion process was nearly completed.

TrachyPen's insertion pattern was similar to the blunt dilator but with a larger mean force of 29.10 N. TrachyPen needs to contact and press the surface skin until the dilator arms successfully enter the stoma, during which all TrachyPen trials experienced a significant increase in the insertion force up to 70.6 N. The insertion force rapidly reduced below 20 N once the dilator arms' tip penetrated through the skin, which mostly occurred at the end of TrachyPen insertion.

The force profiles and stoma patterns of TrachyPen insertion indicate potential improvements regarding the mechanical design. The reason for these types of force profiles is the shape of the dilator arm tip. Unlike the blunt dilator with a small, sharp tip, the dilator arm has a larger diameter of 3 mm than the blunt dilator and a curved tip, making it more difficult to exert sufficient stress to enable penetration. Excessive tissue compression increases the insertion force and brings extra risks when major penetrations suddenly occur. Therefore, the geometry of the dilator arm tip should be revised for easier insertion with sufficient material strength to withstand the dilation. A possible improvement is to adopt the cone shape when redesigning the dilator arms to reduce the tip size and make it sharper.

The characteristic dilation force of TrachyPen could be identified from the first dilation actuated by the motor. Since a significant decrease of the mean motor load peak was identified between the first and second dilation ($P < 0.001$), which indicates the reduction of tissue tension, one dilation should be sufficient during an actual PDT to create a stoma with an appropriate size for tracheostomy tube insertion. The force applied on the dilator arms during the first dilation was larger than CBR and larger than the dilation force reported by the literature [13]. Possible reasons are:

- The porcine back skin is significantly thicker than the human neck, which could increase the difficulty of insertion.
- Different types of PDT instruments (such as CBR, GWDF, and Percutwist) may require different amounts of force to expand the stoma due to their geometries and mechanisms.

The variation of insertion and dilation force data for both TrachyPen and CBR was observed, which could be attributed to the following reasons:

- The thickness difference of the samples could cause a change in insertion force. The thicker the sample, the more difficult it is to penetrate through or expand the stoma.
- The texture and material properties of the porcine back skin samples may vary. Samples were cut from a large area of porcine back, which could have different skin hardness and thickness at different positions.

The uncertainty of the results obtained from the experiments was dominated by the material properties of the porcine tissue, which could be affected by the meat cut, thickness, and moisture condition. No significant deformation of the TrachyPen parts was observed, nor was there a significant deviation from the defined movement trajectory. The uncertainty could be mitigated by selecting the samples from the same fresh porcine back composition with consistent thickness.

Furthermore, the concept of robotic PDT puncture and dilation was proposed along with the experimental setup. TrachyPen has the potential to be used as the end-effector for robotic PDT procedures, which could consist of the following steps:

1. Accurately obtain the puncture position and direction through a sensing technique.
2. The robot is guided to perform a needle puncture and dilation with TrachyPen autonomously.
3. Manually insert the tracheostomy tube to complete the procedure.

TrachyPen forms a component of the robotic system for performing PDT; however, there are some limitations regarding the whole system that could be improved. Currently, the robot movement for using TrachyPen and CBR is guided using

predefined pose trajectories and kinematics only, which is impractical for actual procedures with two major barriers: the PDT guidance and implementation of force feedback. Without direct visualization of the trachea, measuring the position and direction of the puncture is essential for accurate and safe PDT procedures. A sensing technique to guide the robotic PDT will be one of the most critical functions of the whole robotic system. For better control of the TrachyPen insertion, the force feedback should be input to the control loop for robot dynamics. For robotic PDT using CBR, impedance control based on the insertion force is necessary for better simulating the operator so that the instrument pose and the applied force could be adjusted on a real-time basis. Utilizing force feedback in the robot control could significantly enhance the safety level of the robotic PDT procedure.

Therefore, the future improvements of this work are summarized as follows:

- Revise the mechanical design of TrachyPen based on the current force profile and obtain more clinical feedback during the design stage.
- Implement the sensing technique for accurately obtaining the PDT puncture pose and guide the robotic procedure.
- Implement force control and other advanced control algorithms to achieve precise and safe PDT procedures.
- Conduct the tissue damage tests to evaluate the damage caused by the TrachyPen mechanism using a histological approach.

## Conclusions

This research proposes a new prototype medical device for performing PDT needle puncture, blunt insertion, and dilation consecutively. Either by manual holding or being installed on a robotic platform, TrachyPen is designed to combine several PDT procedures into a single step, which eliminates the use of multiple instruments and their complex operations during PDT. Experiments done on the porcine back skin by a robot manipulator have verified the feasibility of performing PDT needle punctures and dilations using TrachyPen. A smaller insertion force profile and stoma length have been identified for using TrachyPen compared with the frequently used dilator (CBR) in clinical practice. TrachyPen has demonstrated the potential feasibility of being used for performing PDT procedures robotically without applying excessive stoma or force. The future work will focus on the TrachyPen design revision and the implementation of other subsystems, such as PDT navigation and robot control, which could finally lead to the development of a robotic autonomous PDT procedure.

## Supporting information

**S1 File. Ethical exemption.**
(PDF)

**S2 Dataset. Trachy data.**
(XLSX)

## Acknowledgments

The authors would like to thank the clinicians from Wythenshawe Hospital and the technical staff from the University of Manchester manufacturing workshop for their support in developing and evaluating the prototype.

## Author contributions

**Conceptualization:** Yuan Tang.

**Formal analysis:** Yuan Tang, Glen Cooper, Brendan A. McGrath.

**Investigation:** Yuan Tang.

**Methodology:** Yuan Tang, Glen Cooper, Damian Crosby, Brendan A. McGrath, Andrew Weightman.

**Software:** Yuan Tang.

**Supervision:** Brendan A. McGrath, Andrew Weightman.

**Validation:** Yuan Tang, Glen Cooper, Brendan A. McGrath.

**Writing – original draft:** Yuan Tang.

**Writing – review & editing:** Yuan Tang, Glen Cooper, Damian Crosby, Brendan A. McGrath, Andrew Weightman.

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
