## [Decision Letter · Decision Letter 0]

1 Oct 2025

PONE-D-25-25649Design and evaluation of a prototype medical device for robotic and manual percutaneous dilatational tracheostomyPLOS ONE

Dear Dr. Tang,

Thank you for submitting your manuscript to PLOS ONE. After careful consideration, we feel that it has merit but does not fully meet PLOS ONE’s publication criteria as it currently stands. Therefore, we invite you to submit a revised version of the manuscript that addresses the points raised during the review process.

We look forward to receiving your revised manuscript.

Kind regards,

Shigao Huang

Academic Editor

PLOS ONE

**Journal Requirements:**

1. When submitting your revision, we need you to address these additional requirements. Please ensure that your manuscript meets PLOS ONE's style requirements, including those for file naming. The PLOS ONE style templates can be found at https://journals.plos.org/plosone/s/file?id=wjVg/PLOSOne_formatting_sample_main_body.pdf and https://journals.plos.org/plosone/s/file?id=ba62/PLOSOne_formatting_sample_title_authors_affiliations.pdf 2. Please update your submission to use the PLOS LaTeX template. The template and more information on our requirements for LaTeX submissions can be found at http://journals.plos.org/plosone/s/latex. 3. We are unable to open your Supporting Information file “Trachy data2.sav”. Please kindly revise as necessary and re-upload. 4. Your ethics statement should only appear in the Methods section of your manuscript. If your ethics statement is written in any section besides the Methods, please move it to the Methods section and delete it from any other section. Please ensure that your ethics statement is included in your manuscript, as the ethics statement entered into the online submission form will not be published alongside your manuscript. 5. Please include captions for your Supporting Information files at the end of your manuscript, and update any in-text citations to match accordingly. Please see our Supporting Information guidelines for more information: http://journals.plos.org/plosone/s/supporting-information. 6. If the reviewer comments include a recommendation to cite specific previously published works, please review and evaluate these publications to determine whether they are relevant and should be cited. There is no requirement to cite these works unless the editor has indicated otherwise.

Reviewers' comments:

Reviewer's Responses to Questions

**Comments to the Author**

1. Is the manuscript technically sound, and do the data support the conclusions?

Reviewer #1: Yes

Reviewer #2: Yes

2. Has the statistical analysis been performed appropriately and rigorously?

Reviewer #1: Yes

Reviewer #2: Yes

3. Have the authors made all data underlying the findings in their manuscript fully available?

Reviewer #1: Yes

Reviewer #2: Yes

4. Is the manuscript presented in an intelligible fashion and written in standard English?

Reviewer #1: Yes

Reviewer #2: Yes

5. Review Comments to the Author

**Reviewer #1:** This manuscript is highly creative and ambitious, offering content that is both engaging and thought-provoking.

With minor refinements in readability and stricter adherence to the journal’s submission guidelines, it has the potential to become an outstanding contribution.

I would like to praise this study for its challenging scope and engaging nature.

**Reviewer #2:** The work is very interesting and is the need of hour. The present research is original in terms of the dilator design and mechanism. A qualitative performance comparison of the dilator design with that of the existing PDT device was done and presented and appears satisfactorily and is novel.

Comparison of the mean insertion force peak with that of the other device available in the market shows the reduction in the force which is a good pointer to pitch for this device though the reduction is not significant. Notwithstanding this the proposed PDT device may provide further insights for future improvement in the device’s design in reducing the pain further and negating other procedural difficulties.

The article complies with the standards of the journal and also in terms of reporting guidelines etc. The content, results and discussion, analyses, presentation and language are good.

Following suggestions may help in the completeness of the manuscript:

1. Uncertainty in the obtained results be presented for better understanding of the performance of the device.

6. PLOS authors have the option to publish the peer review history of their article (what does this mean?). If published, this will include your full peer review and any attached files.

Reviewer #1: No

Reviewer #2: No

---

## [Author Response · Author response to Decision Letter 1]

6 Oct 2025

We thank both reviewers and the editors for the time taken to provide thoughtful reviews of our manuscript. We have responded to reviewers' comments to the best of our abilities, and we hope the paper is now acceptable for publication. We have attached a response letter and a manuscript with track changes.

---

## [Decision Letter · Decision Letter 1]

2 Feb 2026

Design and evaluation of a prototype medical device for robotic and manual percutaneous dilatational tracheostomy

PONE-D-25-25649R1

Dear Dr. Tang,

We’re pleased to inform you that your manuscript has been judged scientifically suitable for publication and will be formally accepted for publication once it meets all outstanding technical requirements.

Kind regards,

Paulo Jorge Simões Coelho

Academic Editor

PLOS One

Additional Editor Comments (optional):

Dear authors, we are pleased to verify that you meet the reviewer's valuable feedback to improve your research.

Thank you for considering PLOS ONE.

Kind regards

PCoelho

Reviewers' comments:

Reviewer's Responses to Questions

**Comments to the Author**

1. If the authors have adequately addressed your comments raised in a previous round of review and you feel that this manuscript is now acceptable for publication, you may indicate that here to bypass the “Comments to the Author” section, enter your conflict of interest statement in the “Confidential to Editor” section, and submit your "Accept" recommendation.

Reviewer #1: All comments have been addressed

Reviewer #3: All comments have been addressed

2. Is the manuscript technically sound, and do the data support the conclusions?

Reviewer #1: Yes

Reviewer #3: Yes

3. Has the statistical analysis been performed appropriately and rigorously?

Reviewer #1: Yes

Reviewer #3: Yes

4. Have the authors made all data underlying the findings in their manuscript fully available?

Reviewer #1: Yes

Reviewer #3: Yes

5. Is the manuscript presented in an intelligible fashion and written in standard English?

Reviewer #1: Yes

Reviewer #3: Yes

6. Review Comments to the Author

Reviewer #1: I would like to express my highest respect for your dedication and commitment.

his is an exceptionally creative and commendable approach

Reviewer #3: All the comments provided by the previous reviewer have been addressed. I have no further comments at this time.

7. PLOS authors have the option to publish the peer review history of their article (what does this mean?). If published, this will include your full peer review and any attached files.

Reviewer #1: No

Reviewer #3: No

---

## [Editor Report · Acceptance letter]

PONE-D-25-25649R1

PLOS One

Dear Dr. Tang,

I'm pleased to inform you that your manuscript has been deemed suitable for publication in PLOS One. Congratulations! Your manuscript is now being handed over to our production team.

Kind regards,

on behalf of

Dr. Paulo Jorge Simões Coelho

Academic Editor

PLOS One